# The Prevalence of Adolescent Social Fears and Social Anxiety Disorder in School Contexts

**DOI:** 10.3390/ijerph191912458

**Published:** 2022-09-30

**Authors:** Francisca Alves, Diana Vieira Figueiredo, Paula Vagos

**Affiliations:** 1Center for Research in Neuropsychology and Cognitive Behavioral Intervention–CINEICC, Faculty of Psychology and Educational Sciences, University of Coimbra, 3000-115 Coimbra, Portugal; 2Institute of Human Development, Portucalense Infante D. Henrique University, 4200-072 Porto, Portugal

**Keywords:** social fears, social anxiety disorder, adolescence, school contexts, prevalence

## Abstract

Social fears arise when fearing to be judged in social events. When these fears are intense, persistent, and debilitating, the individual may suffer from social anxiety disorder (SAD), which has its most frequent onset during adolescence and tends to be chronic. Still, evidence on the prevalence of social fears and SAD in adolescence is scarce. This study analyzed the prevalence of social fears and of SAD in Portuguese adolescents. Of the initial sample (*n* = 1495), 26% presented with intense self-reported social fears. Of those, 53.9% accepted to be further assessed for diagnosis, resulting in a point-estimate prevalence of adolescent SAD of 9.4%; this is slightly higher than previously found. Social performance was the most feared social event. Of the adolescents with SAD, 12.9% were receiving psychological intervention, 12.1% refused intervention, and 92 (65.7%) accepted intervention. Findings confirm SAD as a highly prevalent mental disorder among adolescents, particularly girls, and additionally, that most of these adolescents did not seek treatment but are willing to receive help if made available. Hence, schools should be invested not only in identifying vulnerable adolescents but also in providing diverse intervention options, tailored to their needs, and directing them to successful developmental trajectories.

## 1. Introduction

Social fears refer to anxiety or discomfort in social situations in which one may be and/or anticipates being exposed to evaluation from others, which in turn may result in embarrassment or humiliation. From an evolutionary perspective, social fears are an adaptive experience—they provide a social barometer that contributes to the adjustment of oneself to peers, to the improvement and preservation of social desirability, and to careful awareness of one’s presentation and/or behaviors [1]. However, difficulties may arise when normative social situations are perceived as posing disproportionate threats, and that perception causes a myriad of other symptoms associated with significant impairment. That is the case with social anxiety disorder (SAD), which is at the extreme end of the social anxiety continuum. SAD is characterized by a marked and persistent fear of everyday social and/or performance situations [2]. Individuals with SAD often report concerns such as fear of one’s anxiety symptoms (i.e., shaking, sweating, blushing, stuttering) being visible to others and/or fear of seeming anxious, strange, incompetent, or uninteresting [3]. 

SAD has its most frequent onset during adolescence (i.e., before the age of 18) [4,5,6]. Most of the social situations reported to be difficult by adolescents with SAD occur in school contexts and include talking to peers, starting or joining a conversation, writing on the board, and other social performances tasks (e.g., athletic, reading aloud in class) [7]. When left untreated, difficulties associated with SAD seem to perpetuate into adulthood [8,9] following a typically chronic and debilitating course [10]. SAD can lead to significant impairments and have impact in various domains of the life of adolescents (e.g., greater risk for school dropout, poorer qualifications, increased likelihood to be a victim of bullying, fewer friends, less gratifying interpersonal relationships) [11,12]. Over time, these difficulties are associated with an overall reduction in quality of life [12]. Furthermore, SAD has been linked to increased vulnerability to other anxiety and mood disorders [13,14], behavioral disorders [14], and substance abuse [13,15].

Despite the growing body of research concerning the recognition of, evaluation of, and intervention with SAD, the disorder still seems to be neglected. Specifically, SAD presents one of the lowest rates of treatment seeking [16,17]. This may be associated with several factors: on the one hand, social anxiety seems to be underrecognized and elicit less concern and help-seeking recommendations by adolescents themselves [18]; additionally, social anxiety is perceived to be embarrassing to have, which in turn predicts (alongside other variables) how much others want to distance themselves from the socially anxious person [19]; finally (and paradoxically), experiencing SAD symptoms and being aware of the stigma associated with mental health in general has also been associated with less intention to seek help, particularly from peers and informal adults [20]. This reluctance to ask for support might indicate that the real impact of this disorder is not known, especially in adolescents. Alongside this gap, evidence regarding the prevalence of SAD is scarce, particularly in adolescent samples. Existing research on the prevalence estimates of SAD can be found in Table 1.

It shows that only a minority of studies included referred to adolescent samples [15,21,27,32,38]. Prevalence rates greatly varied, with estimates ranging from 0.8% to 36%. Prevalence estimates may present discrepancies due to geographical and cultural factors [47]. As such, studies conducted in Eastern countries [48] and Western countries [37] showed disparate prevalence rates, concluding that Eastern societies tend to report lower estimated rates of SAD. Additionally, age ranges (i.e., increasing age can lead to higher SAD prevalence; e.g., [42]), methodological differences and/or gender seem relevant variables. Prevalence rates of SAD were systematically found to be higher in females when compared to males, with women also reporting greater clinical severity and higher levels and number of social fears. This gender-sensitive prevalence of SAD seems to be more preponderant in adolescence and become more moderate along the course of life [49]. However, some studies reported an equal gender distribution in adolescence (i.e., statistically non-significant differences between females and males regarding the prevalence of SAD) [14,30]. 

Acknowledging the prevalence of impairing mental disorders such as SAD in a critical developmental phase such as adolescence is an important epidemiological responsibility [50]. Advancing epidemiological understanding of SAD may help to promote preventive and interventional approaches targeted at early ages, who are usually more reluctant to look for specialized treatment [20]. Still, evidence on that prevalence has been scarce recently, particularly so considering adolescent samples only and considering the continuity of social anxiety, going from normative to high and then impacting intensity. As such, the current work will explore the prevalence of heightened levels of social anxiety and of SAD; those presenting with heightened levels of social anxiety may be at higher risk for SAD but are rarely considered in the literature. Because SAD is proposed to be more prevalent in females [49], we expect to replicate that finding in relation to adolescent SAD. Moreover, we expect to confirm previous indications that SAD presents one of the lowest rates of treatment seeking [16,17], but we are also interested in exploring the prevalence of who would be willing to receive treatment if offered, which has not been considered before. Finally, we are interested in exploring the intensity of different core social fears associated with SAD (i.e., observation, social interaction, and social performance). The existence of such core fears is implicit in the diagnostic criteria for SAD, but adolescent SAD has not been explored in relation to those fears. Because these criteria consider a performance specifier for SAD [2], we would expect those fears to be particularly noteworthy. Findings taken from this work may help refine intervention guidelines for those presenting with SAD that are or not proactively seeking treatment. 

## 2. Materials and Methods

### 2.1. Participants

The initial sample consisted of 1495 participants aged between 15 and 18 years old (M = 15.71; SD = 0.82), of which 62.3% were girls (*n* = 932) and 37.7% were boys (*n* = 563). 

### 2.2. Instruments

#### 2.2.1. Screening for Intense Social Fears

The Social Anxiety Scale for Adolescents (SAS-A) [28,51] was used for screening adolescents who might present with heightened social anxiety and could be further assessed for SAD. It is a self-report questionnaire that assesses adolescents’ subjective experiences of social anxiety. It consists of 22 items (e.g., “I worry that others don’t like me”) answered on a 5-point Likert scale according to how much the item “is true for you” (1 = “not at all” to 5 = “all the time”). Scores may range from 22 to 110 and higher scores reflect higher levels of social anxiety. The scale comprises a total score and three subscales, namely the Fear of Negative Evaluation (FNE), the Social Avoidance and Distress of New Situations (SAD-New), and the Generalized Social Avoidance and Distress (SAD-General). In the original study, the FNE, SAD-New, and SAD-General subscales presented good internal consistencies with Cronbach’s alpha values ranging from 0.76 to 0.91 [28]. In the SAS-A Portuguese version, Cronbach’s alpha values reached 0.87 for FNE, 0.74 for SAD-New, 0.71 for SAD-General, and 0.88 for the total score, also representing a good internal consistency [51]. For the screening phase of the current work only the total score was used, and it presented a good internal consistency with a Cronbach’s alpha of 0.89. 

#### 2.2.2. Diagnostic Assessment 

The Mini International Neuropsychiatric Interview for Children and Adolescents (MINI-KID) [22,52] was used for defining the presence/absence of a diagnosis of SAD. It is a structured diagnostic interview for the assessment of DSM-V Axis I diagnoses in children and adolescents. The MINI-KID utilizes a branching tree logic, wherein two to four screening questions are presented for the evaluation of specific diagnostic criteria for each clinical diagnosis. All questions are answered in a yes/no format. If the screening questions are positively answered, additional symptom questions are presented. The MINI-KID includes items that allow the exclusion of medical, organic, and/or drug causes for disorders. Administration time ranges from 30 to 90 min. In its original version, the MINI-KID presented excellent interrater reliability across diagnoses except for dysthymia [22]. The Portuguese version of MINI-KID resulted from a careful translation and backtranslation process and has been previously utilized as a method for diagnoses assessment [52]. Clinicians conducting the assessment phase of the present study received specific training, engaged in role-play training exercises, and observed experienced evaluators applying the interview before conducting individual evaluations.

#### 2.2.3. Self-Reported Core Social Fears

The Social Anxiety and Avoidance Scale for Adolescents (SAASA) [29] was used to characterize the core social fears of adolescents presenting with SAD. It comprises 30 items in its adapted version for late adolescents [53]. It includes two scales assessing the intensity of anxiety and the frequency of avoidance in social situations across the same six factors: interaction with the opposite sex, assertive interaction, observation by others, interaction in new social situations, performance in social situations, and eating and drinking in public. The items are rated on a five-point Likert-type scale (ranging from 1 = “none” to 5 = “very much” for anxiety, and from 1 = “never” to 5 = “almost always” for avoidance). Scores may range from 30 to 150 for each scale and higher scores reflect higher levels of social anxiety/avoidance. In its 30-item version tailored for late adolescents, the SAASA has shown adequate internal consistency values with Cronbach’s alphas over 0.70 [53]. For the present study, a version of the scale that groups the items into three factors related to the main social fears reported by adolescents (i.e., being observed in public, performance, and social interactions [54]) was used for comparing core social fears experienced by adolescents with SAD. Using this sample and this version of the measurement model, good internal consistency was found with Cronbach’s alphas ranging from 0.81 to 0.89. 

### 2.3. Procedures

The sample was collected from Portuguese high schools located in the north and center regions of Portugal. The procedures and goals of the study were shared with thirty-eight schools, following which consent for participating in this work was requested from their executive boards. Twenty-six schools (4 from the north region and 22 from the center region) accepted to collaborate in the implementation of the project. Written and informed consent from the guardian or legal representative was requested for all students attending the 10th and 11th grades. Potential participants were informed on the goals and procedures of the research and the confidentiality and anonymity of their responses were guaranteed. Students were asked to voluntarily participate in the study and informed verbal assent was requested. 

The study then included three phases (i.e., screening for intense social fears, diagnostic assessment, and intervention referral). The screening phase consisted of requesting assenting students to fill in the Portuguese version of the Social Anxiety Scale for Adolescents (SAS-A; [51]). Adolescents who participated in this screening composed the initial sample of this work (see above) that was used to analyze the prevalence of intense self-reported social fears. The following diagnostic assessment phase of this work invited adolescents scoring one standard deviation above the mean found for a large normative sample on the SAS-A (i.e., M = 46.97, SD = 11.16; [51]) for an individual structured clinical interview (i.e., the Portuguese version of the Mini International Neuropsychiatric Interview for Children and Adolescents; [52]). This allowed us to analyze the prevalence of SAD. Based on their diagnostic assessment using the MINI-KID, participants fulfilling inclusion (i.e., primary diagnosis of SAD) and exclusion criteria (i.e., primary diagnosis other than SAD, psychotic symptoms, suicidal risk, special education needs) segued to intervention referral. At this stage, adolescents who were referred to intervention within the research project from which these data were taken filled in a self-report protocol on social fears and other related variables; their scores on social fears were used to characterize diverse core social fears associated with adolescent SAD. Further information on the intervention implementation phase of the project will not be detailed in the present study—for more information contact the principal investigator of the research project and/or ClinicalTrials.gov Identifier: NCT04979676. 

## 3. Results

Figure 1 summarizes prevalence estimates found for the screening, diagnostic assessment, and intervention referral phases of the current work. 

### 3.1. Prevalence of Intense Self-Reported Social Fears

Of our initial sample (*n* = 1495), 26% (*n* = 388) presented with intense self-reported social fears (i.e., scoring one standard deviation above the mean found for a large adolescent normative sample; [20]). There were significant differences between participants reporting and not reporting intense self-reported fears concerning their scores on the complete measure of the SAS-A (M = 66.38, SD = 7.68 and M = 41.69, SD = 9.07; t(1493) = 7.96, *p* < 0.001, *d* = 2.94). 

Of those presenting intense self-reported social fears (*n* = 388), 209 (53.9%) accepted to participate in the diagnostic assessment phase of the study; 179 (46.1%) were not assessed. Those who accepted being assessed reported significantly higher levels of self-reported social anxiety for the complete SAS-A (M = 67.47, SD = 7.50) than those who were not assessed (M = 65.11, SD = 7.71; t(386) = 3.05, *p* = 0.002, *d* = 0.31).

### 3.2. Prevalence of Social Anxiety Disorder

Of adolescents participating in the diagnostic assessment phase of the study (*n* = 209), 140 (67%) presented a diagnosis of SAD, of which 118 (56.5%) had SAD as their primary diagnosis; alternatively, 33% (*n* = 69) did not fulfil diagnostic criteria for SAD. Those who fulfilled (M = 67.64, SD = 7.59) and did not fulfil (M = 67.12, SD = 737) those diagnostic criteria did not differ significantly on self-reported social anxiety based on the complete measure of the SAS-A (t(207) = 0.48, *p* = 0.63, *d* = 0.06). Relating to the initial sample (*n* = 1495), 9.4% of the participants had a SAD diagnosis, of which 7.9% had SAD as their primary diagnosis. Of the 140 adolescents diagnosed with SAD, 76.4% were girls (*n* = 107) and 23.6% were boys (*n* = 33). In relation to the complete sample (*n* = 1495), this signifies a prevalence point-estimate of SAD of 11.16% in girls and 5.86% for boys. Significant differences were found between boys and girls regarding the presence/absence of a SAD diagnosis (χ2(1) = 13.82, *p* < 0.001, *d* = 0.19). The standardized residual value (>|1.9|) shows that there are more girls than boys than statistically expected to have a SAD diagnosis. Alternatively, fewer boys than statistically expected had a SAD diagnosis. Still, though girls scored higher (M = 68.34, SD = 7.55) than boys (M = 65.39, SD = 7.39), the between-gender difference on the complete measure of the SAS-A was not statistically significant (t(138) = 1.97, *p* = 0.05, *d* = 0.39); it was, however, statistically significant for the fear of negative evaluation subscale only (t(138) = 3.04, *p* = 0.003, *d* = 0.56), with girls presenting significantly higher scores (M = 25.39, SD = 3.33) than boys (M = 23.21, SD = 4.30). 

### 3.3. Intervention Referral

Of the adolescents diagnosed with SAD (*n* = 140), 92 (65.7%) were available for segueing to the intervention phase of the study. Alternatively, 17 (12.1%) refused intervention and 13 (9.3%) adolescents did not segue into the intervention phase due to other motives (e.g., special education needs, SAD not being the primary diagnosis). Only 18 (12.9%) were already receiving psychological intervention for SAD. Self-reported social fears, based on the complete SAS-A, did not differ significantly across these groups (F(3139) = 1.81, *p* = 0.15, *f* = 0.21).

The 92 participants who segued to the intervention phase of the study completed the SAASA (along with other measures) at pre-intervention. Using those data, a mixed ANOVA using gender as between-subject and measures as within-subject factors showed that only the main effect of measure was statistically significant (F(2, 180) = 42.11, *p* < 0.001, η2 = 0.32); the main effect of gender (F(1, 90) = 2.57, *p* = 0.11, η2 = 0.028) and the interaction effect (F(2, 180) = 1.99, *p* = 014, η2 = 0.022) were not statistically significant (see Figure 2). Post hoc pairwise comparisons using the Bonferroni correction showed that all three factors differ from each other, and this difference is statistically significant (*p* < 0.001). When the three social fears are compared—being observed (M = 2.69; SD = 1.05), social interactions (M = 3.19; SD = 0.82), and performance situations (M = 3.77; SD = 0.99)—performance situations are highlighted as the most feared by adolescents.

## 4. Discussion

SAD is a debilitating and highly prevalent condition in adolescence and is related to significant impairments in multiple domains of life [12]. Because difficulties associated with this disorder tend to persist throughout life, early detection and intervention are critical research concerns [9]. However, epidemiological information regarding SAD prevalence rates, particularly in adolescence, is still very limited. The present study aimed to bridge this gap by estimating the current prevalence rates of intense self-reported social fears and of SAD as per DSM-V diagnostic criteria, using a representative Portuguese sample. 

About a quarter of adolescents recruited from school contexts reported intense social fears (i.e., 26%), though only about half of them was available to better explore their social fears by participating in a diagnostic interview. This unavailability may reflect the mental health stigma associated with social anxiety in adolescence. Previous evidence has pointed to social anxiety being subjected to perceived/societal stigma [19] and that such perceived stigma had a demotivating effect in help-seeking behavior, even in the presence of SAD symptoms [20]. Because this subthreshold social anxiety has been linked with significant impairment [55], it seems important to be aware of and invest in prevention strategies towards participants that might present risk factors for SAD. For instance, following upon previous evidence on the efficacy of school-based intervention directed at improving general knowledge about mental illness and at promoting help-seeking and help behaviors (e.g., [56,57]), it would be important to specify these interventions to social anxiety in particular, which is particularly unrecognized by adolescents [18].

Of those participants who reported intense self-reported social fears and were willing to be assessed, 33% did not fulfil criteria for a SAD diagnosis based on the DSM-V criteria. Thus, though those participants experienced subjective social anxiety, their experience did not (yet) convey a psychological disturbance. According to previous findings, the trajectory of these participants’ social anxiety symptomatology may depend on several intrapersonal (e.g., behavioral inhibition, [58]; biased interpretation of ambiguous social events, [59]) and interpersonal (e.g., less successful social interactions, [59]) variables. Hence, it would be relevant for school contexts to be aware of these vulnerability profiles, so that adolescents within them may be directed early on to appropriate interventions. 

As for the presence of SAD, our results place the point-estimated prevalence of SAD in Portuguese adolescents at 9.4%; specifically, a ratio of approximately 1 to 3 was found regarding SAD prevalence. This value is slightly higher than data from previous studies that considered similar age groups (e.g., [15,37,38]) and it is much higher than a previous work using a Portuguese adolescent sample [27]. This may reflect the post-confinement period the world is now facing. Specifically, while confined, adolescents might have temporarily felt comfortable and protected from demanding social situations [60]. However, and because social isolation has been generally associated with increased social anxiety-related symptomatology [61], these same adolescents may have felt more prominent social fears upon returning to school contexts [62]. As such, SAD may be a more pressing concern nowadays. 

This point-estimated prevalence was much higher for girls then for boys. Additionally, girls were significantly more prone to experience pathological levels of social anxiety when compared with boys. This gender difference may have been highlighted by the age group under study, as some studies have shown that gender differences in the prevalence of SAD are greatest among adolescents and seem to diminish along the course of life (see [49] for a review). It is also worth noting that the presentation of self-reported social anxiety was similar across boys and girls, except for fear of negative evaluation. The prospective association between fear of negative evaluation and social anxiety over time has been found for adolescence [63], and so may be one of the mechanisms through which girls are more vulnerable to social anxiety than boys. It is also worth mentioning that adolescents with SAD reported most fearing performance in formal events (such as school presentations), followed by interaction with others and then being observed. On the one hand, this lends support to some youth presenting with a more evident (and perhaps circumscribed) fear of performance events, such as anticipated in the DSM-V specifier for performance SAD only [2]. On the other hand, the prominence of this fear may be related to the age range we considered in this work, in which adolescents are likely starting to focus on school achievement as a necessity to progress to higher education. Unfortunately, this fear (and social anxiety in general) may be counterproductive to achieving desired academic goals (e.g., [64]).

Of the 140 diagnosed adolescents, only 18 (12.9%) were already receiving treatment. This is in accordance with previous literature stating that the prevalence of adolescents with SAD diagnosed in school contexts that are not under treatment is very high [27,65], perhaps because difficulties associated with social anxiety have less impact on the school dynamics themselves (i.e., socially anxious students typically do not disturb classes) but rather impact mostly on how the individual internalizes their difficulties (i.e., subjective suffering) and achieves academic success. 

Interestingly, when the possibility of receiving treatment was available, high rates of acceptance were found. Specifically, 65.7% of those diagnosed with SAD were willing to be provided with free psychological intervention for SAD. This result is, surprisingly, higher than what was found in previous studies that have showcased reluctance of adolescents with SAD in seeking treatment (e.g., only 24.1% of adolescents with SAD looked for specialized treatment; [16]). It has been stated that doubts about where to obtain treatment and the fear of being negatively evaluated for seeking it are the main reasons why individuals with SAD are reluctant to seek treatment [66]. In this case, however, adolescents did not have to proactively seek treatment; rather, it was offered to them. This suggests that if treatment is made available for adolescents to accept it (instead of waiting for them to seek it), adherence results will improve and progress can be achieved in closing the gap between the beginning of symptomatology and treatment seeking (previously stated to be of 15 years; [67]). Hopefully—because there are empirically validated interventions available to be implemented specifically in school contexts (e.g., [68])—the trajectories of social fears and SAD of these adolescents can also be changed. 

The findings of the present study further point to school settings as privileged contexts for identifying and treating mental health issues in young ages. Offering school-based prevention and/or intervention programs constitutes an important mental health responsibility that may help bring efficacious treatments to communities that have been systematically found to be under-diagnosed and lacking access to treatment. It may additionally contribute to increasing literacy in mental health (i.e., in school personnel, parents, caregivers) leading to more accurate and earlier identification and referral of mental health issues, including SAD [18]. Considering the possible advantages that interventions in schools can have, future studies may investigate this from a community-based perspective on schools that foster parents, teachers, students and other school personnel as active ingredients of change [69]. 

Despite the relevance of the present study, some limitations should be considered when interpreting its findings. For example, prevalence results/estimates may greatly vary due to different methodological aspects (e.g., age, period of reference). Following this perspective, results should be compared carefully. Additionally, gender was the only sociodemographic variable assessed in the current study. It might be relevant to explore differences between adolescents within different age groups, different years of education and/or different social economic status in future studies. Even though the school context proved to be a privileged setting for the identification of adolescents with SAD, this does not invalidate that other contexts may be important for a broader detection of the disorder. In addition, the present study considered only schools from the north and center regions of the country. Future studies should aim for a broader recruitment. The selection method can also be considered a limitation, since we did not evaluate the entire sample for diagnostic criteria (i.e., only those who scored high on the SAS-A; [51]). As such, we may have lost teenagers who did not self-report social fears.

## 5. Conclusions

The results of this study corroborate that SAD is a highly prevalent mental health disorder among adolescents and that, though being particularly prevalent in girls, boys’ and girls’ social fears seem to present similarly. Though we were not able to explore other demographic correlates and social backgrounds in relation to prevalence of intense social fears or SAD, their potential impact should not be overlooked and that should be considered in future research. Given the major impact SAD holds not only when difficulties emerge but also throughout life when left untreated [10], current findings emphasize the importance of early diagnosis, prevention, and intervention for adolescent social anxiety, which may be particularly useful if adopting a broadened perspective. Specifically, on the one hand, it may be useful to consider not only those adolescents presenting with SAD but also those presenting with intense social fears, and how they may benefit from similar or specific intervention approaches. Cognitive therapy was found to be effective in adult SAD [70] as well as promising in adolescent SAD, whether when delivered in person [71,72] or remotely [73]. Other approaches are also showing potential and should be further explored (e.g., acceptance and commitment therapy [74]). In addition to studying their therapeutic efficacy, it might be important to consider the optimal length of treatment for different intensities of social anxiety symptoms and by that means have different preventive and adequate cost–benefit approaches to adolescent social anxiety. It may be the case that adolescents with intense social fears, compared to those with SAD, are less reluctant to seek help and/or may require less intervention sessions to achieve relevant therapeutic gains; there is no previous evidence on this at this time. On the other hand, schools were an evident context to identify and scrutinize social anxiety and thus should be seen as holistic contexts to be made more aware of how they can help identify vulnerable students and help them cope, in addition to directing them to available interventions. Previous findings have shown that the most frequent recommendation made by adolescents to someone experiencing social anxiety is to seek help from friends [20]. Friends may, then, have a dual role towards their socially anxious peers: they can help their peers appropriately cope with social events if they have sufficient literacy about the difficulties associated with social anxiety, and/or they can be an optimal (albeit indirect) route for socially anxious adolescents seeking formal help. Moreover, because our findings indicate that adolescents with SAD seem to be receptive to intervention being offered to them, school personnel should also be made more aware of the symptoms and suffering associated with social anxiety, so that they can make help-seeking options openly available to those who may benefit from them. Overall, the impact of schools’ mental health literacy on help-seeking, helping and/or being open to treatment could be the focus of future work. Regardless, this inclusive and community-based perspective may overall be the route to prompting higher levels of social functioning, better interpersonal relationships, increased school satisfaction and success on the part of vulnerable adolescents, and consequently, better overall mental health. 

## Figures and Tables

**Figure 1 ijerph-19-12458-f001:**
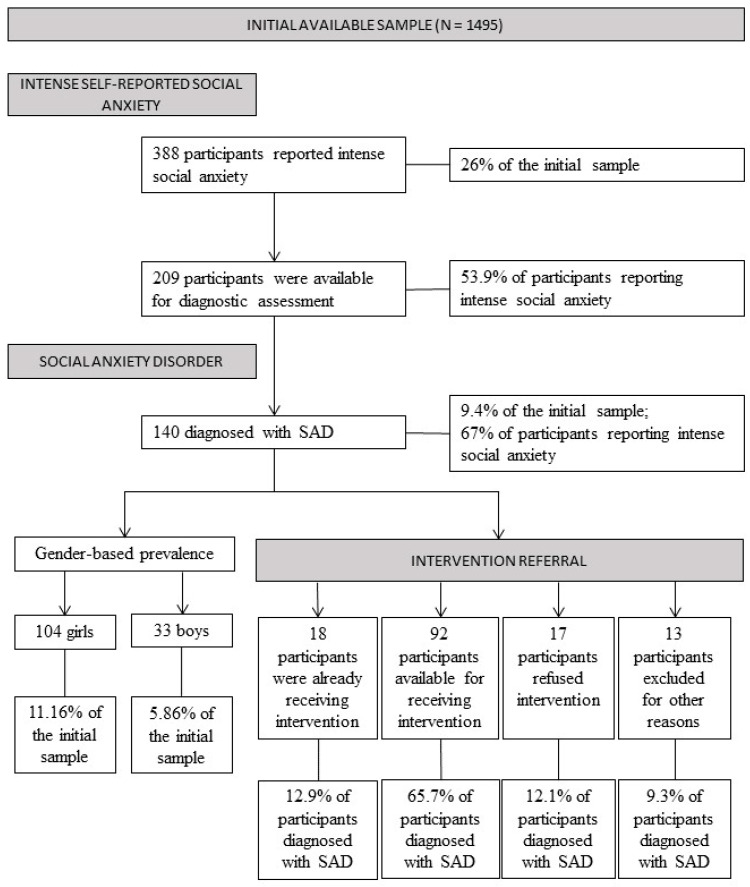
Flowchart of the prevalence estimates found for the screening, diagnostic assessment, and intervention referral phases of the current work.

**Figure 2 ijerph-19-12458-f002:**
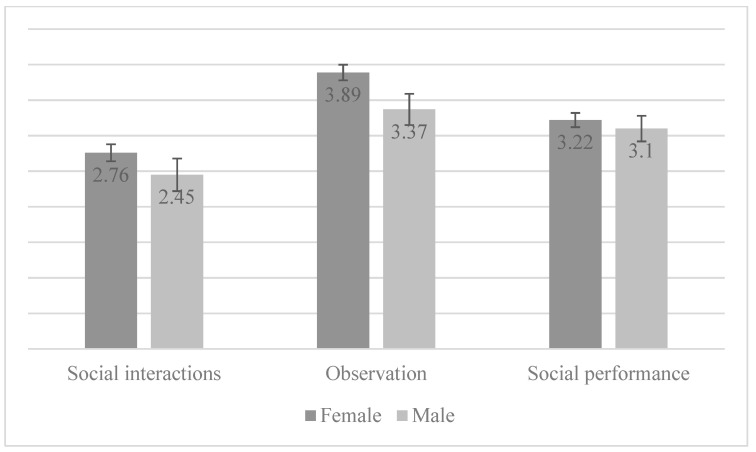
Core social fears by gender.

**Table 1 ijerph-19-12458-t001:** SAD prevalence estimates from existing studies.

Study	Country	Age	Diagnosis Method	Prevalence	Prevalence by Gender
[21]	Saudi Arabia	11–17	MINI-KID [22] (structured interview of DSM-IV diagnoses)	4.6% (time-point)	NR
[23]	Qatar	6–18	Systematic review of the symptoms performed by psychiatrists	12.7% (time-point)	Females (21.9%) had higher rates of all phobic disorders than males (16.8%)
[15]	United States	13–18	CIDI [24] (structured interview of DSM-IV diagnoses)	8.6% (lifetime)	Female (9.2%) Male (7.9%)
[25]	Spain	8–12	SCARED [26] (cut-off point for detecting anxiety disorders symptoms was 25); MINI-KID [22] (structured interview of DSM-IV diagnoses)	3.4% (time-point)	Female (5.5%) Male (2.4%)
[27]	Portugal	12–18	SAS-A [28] (the higher the score, the higher the level of measured anxiety); SAASA [29] (the higher the score, the higher the level of measured anxiety)	3.6% (time-point)	Significant differences were found between girls and boys in all SA indices, with girls reporting higher values. Regarding the number of social fears, 59.4% females reported one or more, while 43.9% of boys reported one or more.
[30]	Iran	6–18	K-SADS-PL [31] (semi-structured diagnostic interview of DSM-IV diagnoses)	0.8% (time-point)	NR
[32]	Sweden	12–14	SPSQ-C [33] (rate at least one potentially phobic situation as marked fear on the social fear scale and it had to be consistently endorsed in the diagnostic questions covering SAD criteria)	4.4% (time-point)	Females (6.6%) Males (1.8%)
[34]	Israel	18–25	LSAS [35] (fear was defined if at least one of the three questions was endorsed)	4.5% (time-point)	NR
[3]	Brazil China Indonesia Russia Thailand United States Vietnam	16–29	SIAS [36] (cut-off point for detecting SAD was 29)	36% (time-point)	Male (35.6%) Female (36.5%) NS
[37]	United States	14–24	CIDI [24] (structured interview of DSM-IV diagnoses)	6.6% (time-period)	NR
[38]	United States	13–18	CIDI [24] (structured interview of DSM-IV diagnoses)	9.1% (lifetime)	Females (12.2%) Males (7%)
[14]	Iran	6–18	K-SADS-PL [31] (semi-structure psychiatric interview based on DSM-IV); MCMI-III [39] (self-administrated psychological instrument)	1.8% (lifetime)	NR
[40]	United Kingdom Germany Italy Portugal Spain	15–101 (exception of Portugal, where the minimum age was set at 18)	Sleep-EVAL Expert System [41] (software with standard questionnaire and diagnostic pathways covering the DSM-IV)	4.4% (time-point)	Female (5.4%) Male (3.4%)
[42]	Australia	4–17	DISC-IV [43] (criteria for impairment required either severe impairment in one or more functional domains or at least moderate impairment in two or more domains)	2.3% (time-period)	Male (2.4%) Female (2.2%) NS
[44]	Chile	4–18	DISC-IV [43] (criteria for impairment required at least two intermediate or one severe criteria) for 4–11 years old; 12–18 years old were directly interviewed	Total = 3.7% 4–11 = 3.5% 12–18 = 3.9% (time-point)	Male (1.8%) Female (5.7%)
[45]	South India	10–16	SPIN [46] (a score less than 20 was considered normal, scores 21–30 as mild, 31–40 as moderate, 41–50 as severe, and more than 51 as a very severe social phobia)	22.9% (time-point)	Male (15.2%) Female (30.9%)

Note. MCMI-III = Millon Clinical Multiaxial Inventory, third edition; K-SADS-PL = Kiddie Schedule for Affective Disorders and Schizophrenia for School-Age Children-Present and Lifetime Version; CGAS = Children’s Global Assessment Scale; LOI-CV = Leyton Obsessional Inventory-Child Version; SCARED = Screen for Child Anxiety Related Emotional Disorders; CDI = Children’s Depression Inventory; LSAS = Liebowitz Social Anxiety Scale; SIAS = Social Interaction Anxiety Scale; SPIN = Social Phobia Inventory; SPSQ-C = The Social Phobia Screening Questionnaire for Children; K-SADS-PL = Kiddie Schedule for Affective Disorders and Schizophrenia Present and Lifetime Version; DISC-IV = Diagnostic Interview Schedule for Children version 4; MINI-KID = MINI International Neuropsychiatric Interview for Children and Adolescents; CIDI = Composite International Diagnostic Interview Version 3.0; NS = not significant; NR = not reported.

## Data Availability

Because the research project from which data for this manuscript were taken is still ongoing, data were not, at this time, made publicly available. Data that supported the results reported in the manuscript may be obtained, upon reasonable request, from the corresponding author.

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
