# Peer review of "The Prevalence of Adolescent Social Fears and Social Anxiety Disorder in School Contexts"

_ijerph, 2022, doi:10.3390/ijerph191912458_

Round 1

Reviewer 1 Report

The reviewer is thankful for the opportunity to review the submitted paper entitled “The Prevalence of Adolescent Social Fears and Social Anxiety Disorder in School Contexts.” I examined the submitted paper with immense interest. The paper focused on the problem of social anxiety disorder in school contexts. To do so, they scrutinized the mental health of young boys and girls belonging to the schools in the North and center regions in Portugal. Consequently, they mainly clarified two facts: one is that social anxiety disorder could be observed among young adolescents (especially, girls) in the schools of Portugal; the other is that a large number of them were not cared for appropriately. The reviewer believes that these findings have a significant meaning for social researchers. Additionally, the authors have succeeded in deriving these findings based on careful and appropriate procedures. Therefore, the reviewer believes that the study implemented by the authors has high reliability. Certainly, the submitted paper has some concerns (e.g., it did not sufficiently examine the effects of sociodemographic factors and social backgrounds on social anxiety disorder.) Alternatively, the authors have fairly discussed the limitations of their study in the “Discussion” section. Concludingly, the reviewer believes that their concerns would be overcome in the future studies by the authors. Based on these, the reviewer believes that the submitted paper is ready for publication.

Reviewer 2 Report

I have some minor comments in this paper. Please deal with my review comments. 

1.Please add effect sizes in the statistical analysis part. 

2. I felt that the analysis was not consistent. First, the SAS-A sub-scales were compared only between L191~L210. I think the 2-way (gender and three sub-scales) ANOVA is suitable, not the t-test. Second, the SAASA sub-scales comparison (repeated ANOVA) was performed only between L219~L227. I think this analysis would need to be performed in other parts. the authors need to explain and emphasize why a specific analysis is needed only for a specific part. 

3. I think the authors can visualize the result of ANOVAs.

Reviewer 3 Report

Thank you very much for the opportunity to read and to contribute to the revision of the manuscript entitled “The Prevalence of Adolescent Social Fears and Social Anxiety Disorder in School Contexts”.

The study explores the prevalence of social fears and social anxiety disorders (SAD) in a sample of Portuguese adolescents, underlining the importance of the school context in identifying vulnerable subjects and promoting prevention and intervention programs.

The topic is interesting and particularly relevant to the current historical period, during which the confinement at home and the disruption of daily relationships in presence has amplified, especially among young people, the difficulties associated with social life.

The topic is well described and adequately discussed. Nevertheless, there are some issues that need to be dealt with, in order to further improve the quality of the paper.

Detailed comments on the different Sections of the paper are reported below:

Abstract

The abstract is concise and easy to read.

Introduction and aims

In general, the rationale of the study is convincing.

Specific suggestions are given below:

-      I suggest a more in-depth discussion of help-seeking by adolescents who experience social anxiety, as this is a central aspect of the work, also addressed in the Discussion Section.

-      I have some concerns about the use of Table 1 (SAD prevalence estimates from existing studies). I think it would be more useful to put the table in the appendix and to report only a summary of the main results in the Introduction. This would make the text easier to read. I also point out a typo in the table notes: please replace "non" with "not".

-      There is a misprint at line 80: please replace “counties” with “countries”.

-      I suggest to describe more fully, at the end of the Introduction Section, the aims of the work; moreover, it would be useful to refer to theoretical premises to make explicit the hypotheses of the work (they are completely lacking).

-      I suggest better highlighting the novelty aspects of the work.

Materials and Methods

The Materials and Methods Section is well systematized but it is somewhat confusing in the description of the use of instruments. In particular:

-      Authors refer to two different scales to evaluate Social Anxiety:  

1) The Social Anxiety Scale for Adolescents (SAS-A)

2) The Social Anxiety and Avoidance Scale for Adolescents (SAASA)

If I understand correctly, the SAS-A was used for the screening phase, aimed at identifying adolescents with a high level of social anxiety. What about SAASA? Was this instrument used for the intervention phase only? If so, why did the authors choose to use two different measures? It is not clear and needs to be explained.

-      I suggest to add the Range in the paragraphs describing the different measures.

Results

-      In the flowchart represented in Figure 1, the group of 13 (9.3%) adolescents who didn’t segue into the intervention phase for other motives (e.g., special education needs, SAD not being the primary diagnosis) is not reported among “Intervention referral”. I suggest adding this category as well in Figure 1, for completeness.

-      For what concerns “Intervention Referral”, results need to be better explained:

1) which measure was used? I suppose the SAASA scale, but it is not clear;

2) the authors say that a repeated-measures ANOVA was performed: when were the data collected? How many times? At what time interval from each other?

Discussion and Conclusions

The Discussion is well reasoned, in line with the aims of the study and supported by data analysis.

I suggest to extend the Conclusions, deeper discussing the possible implications of the study and highlighting new insights for future research.

Round 2

Reviewer 2 Report

Dear authors

Thank you for your kind revising. I have some minor comments at Results section. 

1. In mixed ANOVA, The authors can show the statistical data (e.g., F, df, p-value, partial squared eta) at non significant results.

2. In order to help the reader's understanding, I think that the result of mixed ANOVA can be visualized at least.

Author Response

Thank you for your kind revising. I have some minor comments at Results section.

  1. In mixed ANOVA, The authors can show the statistical data (e.g., F, df, p-value, partial squared eta) at non-significant results.

The statistical data for the non-significant effect of gender and for the non-significant effect of fear*gender is now presented in the manuscript. That segment of the manuscript now reads: “Using that data, a mixed ANOVA using gender as between-subject and measures as within-subject factors show that only the main effect of measure was statistically significant (F(2, 180) = 42.11, p < .001, Æž2 = .32); the main effect of gender (F(1, 90) = 2.57, p = .11, Æž2 = .028) and the interaction effect (F(2, 180) = 1.99, p = 014, Æž2 = .022) were not statistically significant.”

  1. In order to help the reader's understanding, I think that the result of mixed ANOVA can be visualized at least.

Figure 2 was added to the manuscript, presenting core social fears by gender.